# Mapping and analysis of *Caenorhabditis elegans* transcription factor sequence specificities

**Kamesh Narasimhan[1†], Samuel A Lambert[2†], Ally WH Yang[1†], Jeremy Riddell[3], Sanie Mnaimneh[1], Hong Zheng[1], Mihai Albu[1], Hamed S Najafabadi[1], John S Reece-Hoyes[4], Juan I Fuxman Bass[4], Albertha JM Walhout[4], Matthew T Weirauch[5,6,7]\*, Timothy R Hughes[1,2,8]\***

[1]Donnelly Centre for Cellular and Biomolecular Research, University of Toronto, Toronto, Canada; [2]Department of Molecular Genetics, University of Toronto, Toronto, Canada; [3]Department of Molecular and Cellular Physiology, Systems Biology and Physiology Program, University of Cincinnati, Cincinnati, United States; [4]Program in Systems Biology, University of Massachusetts Medical School, Worcester, United States; [5]Center for Autoimmune Genomics and Etiology, Cincinnati Children's Hospital Medical Center, Cincinnati, United States; [6]Divisions of Biomedical Informatics and Developmental Biology, Cincinnati Children's Hospital Medical Center, Cincinnati, United States; [7]Department of Pediatrics, University of Cincinnati, Cincinnati, United States; [8]Canadian Institutes For Advanced Research, Toronto, Canada

**Abstract** *Caenorhabditis elegans* is a powerful model for studying gene regulation, as it has a compact genome and a wealth of genomic tools. However, identification of regulatory elements has been limited, as DNA-binding motifs are known for only 71 of the estimated 763 sequence-specific transcription factors (TFs). To address this problem, we performed protein binding microarray experiments on representatives of canonical TF families in *C. elegans*, obtaining motifs for 129 TFs. Additionally, we predict motifs for many TFs that have DNA-binding domains similar to those already characterized, increasing coverage of binding specificities to 292 *C. elegans* TFs (~40%). These data highlight the diversification of binding motifs for the nuclear hormone receptor and C2H2 zinc finger families and reveal unexpected diversity of motifs for T-box and DM families. Motif enrichment in promoters of functionally related genes is consistent with known biology and also identifies putative regulatory roles for unstudied TFs.

**\*For correspondence:** Matthew. Weirauch@cchmc.org (MTW); t.hughes@utoronto.ca (TRH)

†These authors contributed equally to this work

**Competing interests:** The authors declare that no competing interests exist.

## Introduction

Transcription factors (TFs) are sequence-specific DNA-binding proteins that control gene expression, often regulating specific biological processes such as pluripotency and differentiation (*Takahashi and Yamanaka, 2006*), tissue patterning (*Lemons and McGinnis, 2006*), the cell cycle (*Evan et al., 1994*), metabolic pathways (*Blanchet et al., 2011*), and responses to environmental stimuli (*Benizri et al., 2008*). The nematode *Caenorhabditis elegans* is a powerful model for studying gene regulation as it is a complex and motile animal, yet has a compact genome (~100 Mbp) (*C. elegans Sequencing Consortium, 1998*) featuring relatively short intergenic regions (mean 1389 bp; median 662 bp). Indeed, the observation that proximal promoter sequence is often sufficient to produce complex tissue-specific gene expression patterns (*Dupuy et al., 2004*; *Zhao et al., 2007*; *Grove et al., 2009*; *Sleumer et al., 2009*; *Niu et al., 2011*) indicates that long-range gene regulation

**eLife digest** Many scientists use 'model' species—such as the fruit fly or a nematode worm called *Caenorhabditis elegans*—in their research because these organisms have useful features that make it easier to carry out many experiments. For example, *C. elegans* has a smaller genome compared to many other animals, which is useful for studying the roles of individual genes or stretches of DNA.

Transcription factors are a type of protein that can bind to specific stretches of DNA and help to switch certain genes on or off. These 'motifs' may be close to the gene or further away in the genome, and therefore, must stand out amongst the rest of the DNA, like lights on a landing strip. However, the motifs for only 10% of the estimated 763 transcription factors in *C. elegans* have been identified so far.

In this study, Narasimhan, Lambert, Yang et al. used a technique called a 'protein binding microarray' to identify the motifs for many more of the *C. elegans* transcription factors. These findings were then used to predict motifs for other transcription factors. Together, these methods increased the proportion of *C. elegans* transcription factors with known DNA-binding motifs from 10% to around 40%.

Now that more DNA motifs have been identified, it is possible to look for similarities and differences between them. For example, Narasimhan, Lambert, Yang et al. found that transcription factors with similar sequences can bind to very varied motifs. On the other hand, some transcription factors that are very different are able to recognize very similar motifs. The experiments also indicate that motifs found very close to genes—in sequences known as 'promoters'—may be able to interact with many proteins to influence the activity of genes.

Narasimhan, Lambert, Yang et al.'s findings increase the number of *C. elegans* transcription factors with a motif, bringing the knowledge of these proteins more in line with the better-studied transcription factors of humans and fruit flies. The next challenge is to identify DNA motifs for the remaining 60% of transcription factors.

through enhancers is not as abundant in *C. elegans* as it is in flies or mammals (*Gaudet and McGhee, 2010*; *Reinke et al., 2013*).

*C. elegans* has 934 annotated TFs (*Reece-Hoyes et al., 2005*), and 744 proteins that possess a well-characterized sequence-specific DNA-binding domain (DBD) (*Weirauch and Hughes, 2011*; *Weirauch et al., 2014*). *C. elegans* contains major expansions of several specific TF families, with nuclear hormone receptor (NHR), $Cys_2His_2$ (C2H2) zinc finger, homeodomain, bHLH, bZIP, and T-box together comprising 74% of the TF repertoire (*Reece-Hoyes et al., 2005*; *Haerty et al., 2008*). The lineage-specific expansion of C2H2 zinc finger (ZF) TFs is similar to that observed in many animals, including diversification of DNA-contacting 'specificity residues', suggesting diversification in DNA-binding specificity (*Stubbs et al., 2011*). The *C. elegans* genome encodes an unusually large number of NHRs (274 members), more than five times the number in human (48 members) (*Enmark and Gustafsson, 2001*; *Reece-Hoyes et al., 2005*). It is speculated that the NHRs may serve as environmental sensors (*Enmark and Gustafsson, 2001*; *Arda et al., 2010*), providing a possible explanation for their variety and numbers. Five of the six major NHR sub-families found across metazoa are also found in *C. elegans* (NR3 is lacking), but the vast majority of *C. elegans* NHRs define novel sub-families that are not present in other metazoans (*Van Gilst et al., 2002*) and which are derived from an ancestral gene most closely resembling HNF4 (aka NR2A) (*Robinson-Rechavi et al., 2005*). Extensive variation in the DNA-contacting recognition helix (RH) or 'P-box' suggests that *C. elegans* NHRs, like C2H2 and bHLH families, have diversified DNA-sequence specificities, and that many will recognize novel motifs (*Van Gilst et al., 2002*). The T-box gene family presents another example of a nematode-specific expansion, with 22 members in *C. elegans*, of which 18 lack one-to-one orthologs in other metazoan lineages (*Minguillon and Logan, 2003*). Only four have known binding motifs, and unlike most other TFs, T-box binding motifs are virtually identical across the metazoa (*Sebé-Pedrós et al., 2013*; *Weirauch et al., 2014*); the diversification of TFs is often associated not only with changes in DNA-sequence specificity but also alteration in protein–protein interactions and expression of the TF gene itself (*Grove et al., 2009*; *Reece-Hoyes et al., 2013*).

Despite extensive study of gene regulation, including several large-scale efforts (*Deplancke et al., 2006*; *Grove et al., 2009*; *Lesch et al., 2009*; *Gerstein et al., 2010*; *Niu et al., 2011*; *Sarov et al., 2012*; *Reece-Hoyes et al., 2013*; *Araya et al., 2014*), the landscape of *C. elegans* TF-sequence specificities remains largely unknown. To our knowledge, motifs are currently known for only 71 *C. elegans* TFs, including those determined in single-gene studies, previous protein binding microarray (PBM) analyses, and modENCODE TF ChIP-seq data (*Matys et al., 2006*; *Araya et al., 2014*; *Mathelier et al., 2014*; *Weirauch et al., 2014*). It has been surprisingly difficult to obtain motifs from ChIP-seq data (*Niu et al., 2011*; *Araya et al., 2014*), possibly due to indirect binding, or a dominant role of chromatin structure in either determining in vivo binding sites (*Song et al., 2011*) or in the purification of chromatin fragments (*Teytelman et al., 2013*). Yeast one-hybrid (Y1H) assays (*Reece-Hoyes et al., 2011*) cannot be used easily to derive TF motifs, because the DNA sequences tested are too large (~2 kb on average). However, there is a strong statistical correspondence between motifs determined by PBMs and Y1H data (*Reece-Hoyes et al., 2013*). Computational approaches coupling promoter sequence conservation and/or gene expression data to identify TF motifs de novo have collectively produced many more motifs than there are TFs (*Beer and Tavazoie, 2004*; *Sleumer et al., 2009*; *Zhao et al., 2012*) and also do not inherently reveal the cognate TFs that correspond to each putative motif. Multimeric binding represents one possible complication in the analysis of in vivo TF-binding data (*Ao et al., 2004*). Indeed, TF co-associations were identified based on ChIP-seq peak-binding overlaps in *C. elegans* modENCODE studies (*Araya et al., 2014*), but the underlying sequence recognition mechanisms were not apparent.

Here, we use PBMs to systematically identify *C. elegans* TF DNA-binding motifs. We selected a diverse set of TFs to assay, ultimately obtaining 129 motifs from different TF families and subclasses. The data show that the expansion of most major TF families is associated with diversification of DNA-binding motifs. Motif enrichment in promoters reveals that our motif collection readily associates individual TFs with putative regulated processes and pathways.

## Results

### Overview of the PBM data

The key goal of this project was to expand our knowledge of DNA-sequence specificities of *C. elegans* TFs. To do this, we analyzed a diverse set of TF DBDs (see below) with PBM assays (*Berger et al., 2006*; *Weirauch et al., 2014*). Briefly, the PBM method works by 'hybridizing' a glutathione S-transferase (GST)-tagged DNA-binding protein (in our assays, the DBD of a TF plus 50 flanking amino acids) to an array of ~41,000 defined 35-mer double-stranded DNA probes. The probes are designed such that all 10-mer sequences are present once, and all non-palindromic 8-mers are, thus, present 32 times in difference sequence contexts (palindromic 8-mers occur 16 times). A fluorescently labelled anti-GST antibody illuminates the extent to which each probe is bound by the assayed TF. Using the signal intensity for each probe, the specificity of the TF is derived. For each individual 8-mer, we derive both E-scores (which represent the relative rank of microarray spot intensities, and range from −0.5 to +0.5 [*Berger et al., 2006*]) and Z-scores (which scale approximately with binding affinity [*Badis et al., 2009*]). PBMs also allow derivation of position weight matrices (PWMs) up to 14 bases long (*Berger et al., 2006*; *Mintseris and Eisen, 2006*; *Badis et al., 2009*; *Weirauch et al., 2013*) (hereafter, we take 'motif' to mean PWM). To determine PWMs, we used the data from PBM assays performed on two different array designs to score the performance of PWMs obtained from different algorithms, as previously described (*Weirauch et al., 2013*, *2014*).

In this study, we selected TFs to analyze on the basis of their DBD sequence, aiming to examine at least one TF from each group of paralogous TFs, and biasing against TFs that have known PBM motifs, or close orthologs or paralogs with known motifs (see 'Materials and methods' for full description of selection scheme). The selections were guided by previous PBM analyses that determined sequence identity thresholds for each DBD class that correspond to motif identity (*Weirauch et al., 2014*). To identify TFs, we used the Cis-BP definition of DBDs (*Weirauch et al., 2014*), which employs a list of well-characterized eukaryotic DBDs and a distinct significance threshold for each DBD class. CisBP identified 744 *C. elegans* proteins, encompassing 52 domain types (listed in *Figure 1—source data 1*). 689 (93%) of these 744 are present in the wTF catalog of 934 annotated TFs (*Reece-Hoyes et al., 2005*); thus, these sets are largely overlapping. We manually examined the differences between the two TF lists (see *Supplementary file 1*) and found that most of them can be accounted for by

(i) changes to the *C. elegans* protein catalog over time, (ii) differences in domain classes included, (iii) differences in domain score threshold, (iv) fewer manual annotations in Cis-BP, and (v) ambiguity in classifying C2H2 zinc fingers as TFs. Overall, wTF2.0 contains only 19 proteins that are not in Cis-BP and that are very likely *bona fide* sequence-specific TFs. wTF2.0 also contains 83 C2H2 proteins that fall below the CisBP score threshold, 52 of which have only a single C2H2 domain. DNA recognition typically requires multiple C2H2 domains; however, some fungal TFs do bind DNA with a single C2H2, employing additional structural elements (*Wolfe et al., 2000*). Thus, these proteins have an ambiguous status. In general, Cis-BP excludes proteins with lower domain scores and those with little or no evidence for sequence-specific DNA binding, and we, therefore, refer to the 744 in CisBP plus the 19 additional *bona fide* TFs as the 763 'high confidence' *C. elegans* TFs.

We attempted to clone DBDs from 552 unique high confidence TFs, ultimately obtaining clones for 449, all of which we assayed by PBMs. After employing stringent success criteria (see 'Materials and methods') we obtained sequence-specificity data (8-mer scores and motifs) for 129 DBDs. PBM 'failures' may be due to any of several causes, including protein misfolding, requirement for cofactors or protein modifications (e.g., phosphorylation), or *bona fide* lack of sequence-specific DNA-binding activity. The overall success rate (29%) is comparable to that we have observed from the analysis of thousands of DBDs from diverse species (35%) (*Weirauch et al., 2014*).

A summary of our results is presented in *Figure 1*, broken down by motif numbers and percent coverage for individual DBD classes. Our motif collection encompasses 26 different DBD classes, and greatly increases the number and proportion of *C. elegans* TFs for which motifs have been identified experimentally, from 71 (10%) to 195 (26%) (five of the 129 had previously-known motifs). The new data encompass all of the large TF families, including C2H2 zinc fingers, NHRs, bZIPs, homeodomains, DM domains, and GATA proteins.

## Validation of motifs, motif novelty, and motifs predicted using homology

We next asked whether our new data are consistent with previous knowledge. Of the 129 TFs, only five have previously known motifs, all of which we recapitulated (*Figure 1—figure supplement 1*). The sequence preferences for most of the 129 TFs were different from those of any previously assayed TF, however. The boxplots in *Figure 2A–C* and *Figure 2—figure supplements 1–4* show that, on average, the new TFs we analyzed bound a set of 8-mers that was largely non-overlapping with that of the most similar protein that had been analyzed previously by PBM (red circles indicate the 8-mer overlap between individual TFs analyzed by PBM in our study, and the most similar TF analyzed by PBM in any study). Nonetheless, some pairs of TFs have DBDs that are highly similar and bind highly overlapping 8-mers. These observations are quantitatively consistent with the prior study we used for guidance in selecting TFs (black box plots) (*Weirauch et al., 2014*), and thus, we expect that the scheme for predicting sequence specificity via amino acid identity that was proposed in the prior study can also be used in *C. elegans*. In this scheme, TFs without DNA-binding data are simply assigned the motifs and 8-mer data for other TFs with DBD amino acid similarity above a threshold, if those data exist. These TFs can be from *C. elegans* or from other species. If we include these predicted motifs, then the number of *C. elegans* TFs with an associated motif increases to 292 (39%), including TFs with motifs predicted from other *C. elegans* TFs (24) and those with motifs predicted from other species (79).

## Expert curation of motifs

The entire *C. elegans* motif collection, including our new data, previously published motifs, and those predicted by homology from other TFs in *C. elegans* and other species, encompasses 1769 unique motifs representing only 292 TFs. About half (157, or 54%) of the 292 TFs with motifs are represented by only a single motif, as there were no data prior to our study for these TFs or their close homologs. Some TFs (e.g., homeodomains, PAX, and forkheads), however, are highly conserved and thus have many orthologs above the prediction threshold. In addition, TFs that are known developmental regulators tend to be well studied, and often possess multiple associated motifs. To gain an overview of the full motif collection and to compare among the multiple motifs for each protein, we used the PWMclus tool (*Jiang and Singh, 2014*), with default settings, to obtain groups of highly related motifs from all TFs within each DBD class. This tool uses an information-content weighted Pearson correlation between aligned PWM columns as a similarity

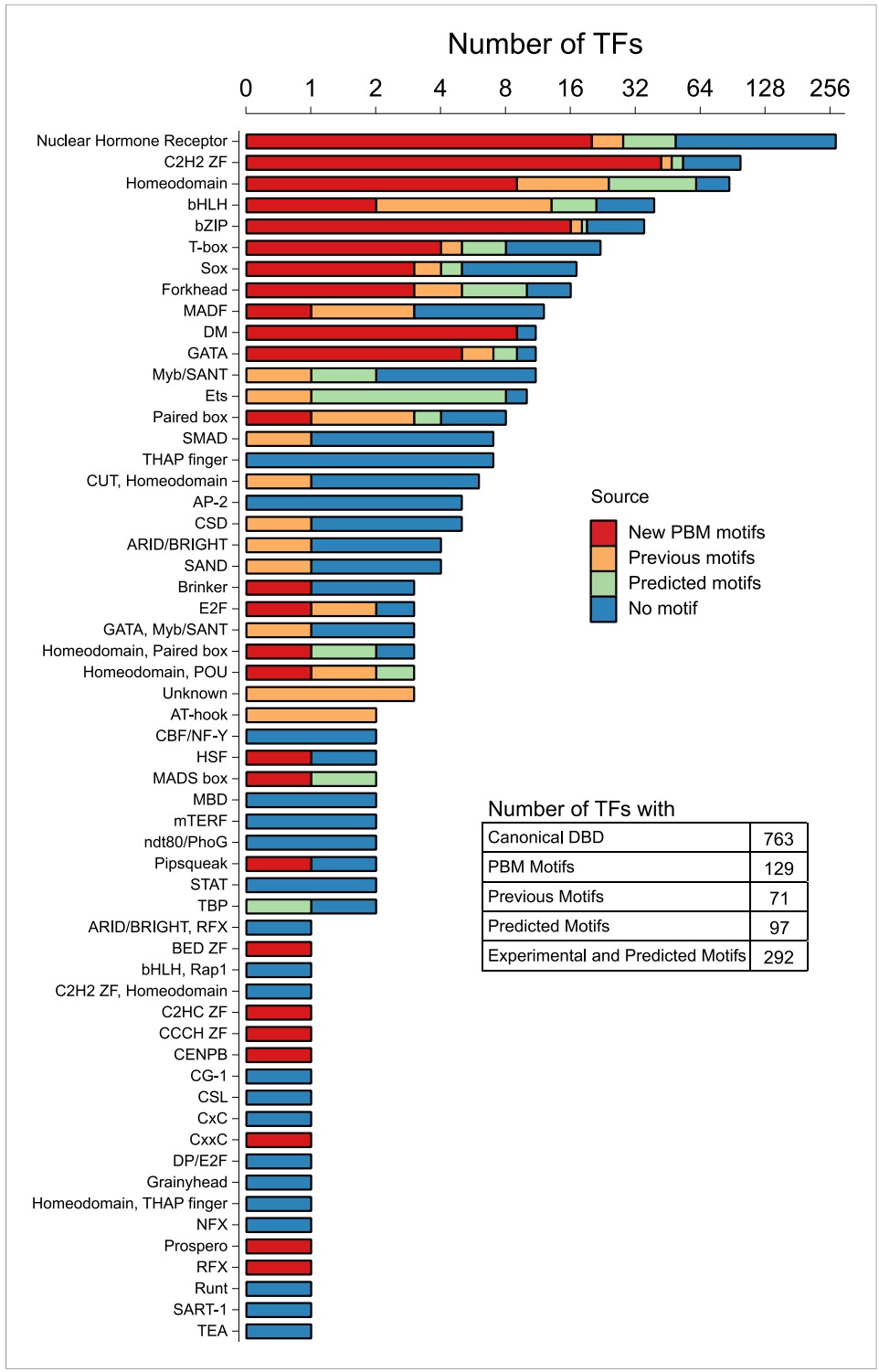

**Figure 1**. Motif status by DBD class. Stacked bar plot depicting the number of unique *C. elegans* Transcription factors (TFs) for which a motif has been derived using PBM (this study), previous literature (including PBMs), or by homology-based prediction rules (see main text). The y-axis is displayed on a $\log_2$ scale for values greater than zero. See *Figure 1—source data 1* for DNA-binding domain (DBD) abbreviations. Correspondence between motifs identified in current study and previously reported motifs is shown in *Figure 1—figure supplement 1*.

The following source data and figure supplement are available for figure 1:

*Figure 1. continued on next page*

*Figure 1. Continued*

**Source data 1**. Table of *C. elegans* TF repertoire motif coverage and list of TF DBDs present in *C. elegans*.

**Figure supplement 1**. Correspondence between TF motifs identified from our PBM study and previously reported motifs from several types of experimental data.

measure for hierarchical clustering, and then selects branches within which the average internal correlation exceeds R ≥ 0.8. This procedure collapsed the 1769 motifs into a set of 424 clusters. This number is still larger than the number of TFs with either known or predicted motifs (292), since there are many cases in which motifs for a single TF are distributed across multiple clusters, although in 67% of cases in which there are multiple known and predicted motifs for a given protein, the majority of them do form a single cluster.

There appear to be several explanations for this phenomenon, as exemplified by the bZIP family shown in *Figure 2D*. First, different studies and different experimental (or computational) techniques often yield motifs for the same protein that are clearly related by visual examination, but score as different from each other using PWMclus. For example, there are four different motifs for SKN-1 (from PBM, Chip-seq, and Transfac) that all contain the same half-site, ATGA, but have different flanking sequence preferences. Similarly, for Forkhead TFs FKH-1 and UNC-130, different methods produce variants with differences in the sequences flanking the core TGTTT Forkhead binding site. A related explanation is that a single motif may not adequately capture all aspects of TF sequence preferences, such as the ability of many TFs to bind as both a monomer and a homodimer (or multimer) with preferred spacing and orientation, variability in the preferred spacing, changes to the preferred monomeric sites that are associated with dimerization, and effects of base stacking that result in preferred polynucleotides at some positions (*Jolma et al., 2013*). In addition, different experimental methods may capture some aspects of DNA-binding complexity better than others.

It is inconvenient to have a large number of motifs for a single protein for several reasons. First, it is difficult to peruse the full motif collection. In addition, comprehensive motif scanning is slower with a large number of motifs, and the motif scans produce partially redundant results that require deconvolution and reduce statistical power. We, therefore, sought to identify a single motif or set of motifs for each protein that are minimally redundant and are best supported by existing data. We used a semi-automated scheme that considers all data available (similar to that described in (*de Boer and Hughes, 2012*); see 'Materials and methods'). Briefly, we prioritized motifs that are (a) measured experimentally, rather than predicted; (b) more similar to other motifs for the same TF, or highly similar TFs, especially if they are derived from in vitro data, which would be free of confounding effects present in vivo; (c) assigned to the cluster that contains the majority of motifs for that TF; (d) most consistent with the type of sequences that a given DBD class typically binds; (e) best supported by ChIP-seq or Y1H data, if available (see below).

This procedure resulted in a set of 284 motifs representing the 292 *C. elegans* TFs with experimentally determined or predicted motifs (*Supplementary file 2*). The outcome for the bZIP family is shown on the right of *Figure 2D*, which illustrates that the motif curation procedure produces motifs that are consistent with known bZIP class binding sites. The curated set also contains 16 cases in which the same protein is represented by multiple motifs (exemplified by the GATA family TF ELT-1, which binds as both a monomer and a homodimer, *Figure 2—figure supplement 5*), and 11 cases in which more than one protein is represented by the same motif (e.g., GATA family TFs MED-1 and MED-2, *Figure 2—figure supplement 5*; in all of these cases, the TFs are highly similar proteins). We also note that PWMclus subdivides the 284 curated motifs into only 127 different clusters (data not shown), because the motif(s) contained in many of the 424 original clusters met few or none of the selection criteria above.

## Overview of PBM 8-mer data

The majority of the expert curated motifs (237, or 84%) are derived from the PBM data described in this study or from previous studies (compiled in [*Weirauch et al., 2014*]), which are the only data

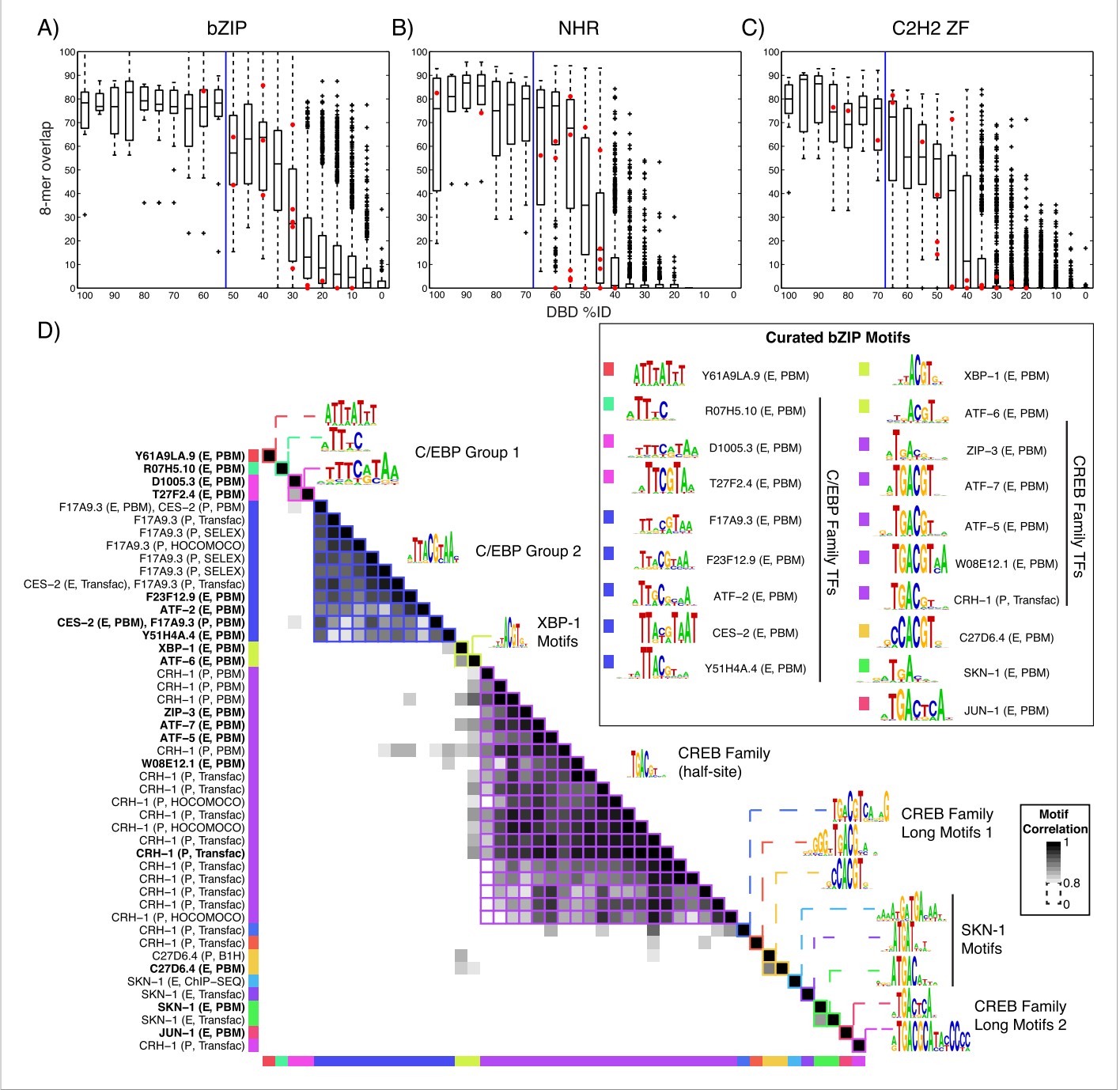

**Figure 2**. Motif prediction, motif clustering, and identification of representative motifs. (**A–C**) Boxplots depict the relationship between the %ID of aligned AAs and % of shared 8-mer DNA sequences with E-scores exceeding 0.45, for the three DBD classes, as indicated. %ID bins range from 0 to 100, of size 10, in increments of five. Red dots indicate individual TFs in this study, vs the next closest TF with PBM data. Vertical lines indicate AA %ID threshold above which motifs can be predicted using homology, taken from (*Weirauch et al., 2014*). Boxplots for all other DBDs in current study are shown in *Figure 2—figure supplements 1–4*. (**D**) Clustering analysis of motifs of bZIP domains using position-weight matrices (PWM)clus (*Jiang and Singh, 2014*). Colored gridlines indicate clusters. Cluster centroids are shown along the diagonal; expert curated motifs are shown within the box at right. 'E' indicates experimentally determined motifs; 'P' indicates predicted motifs. Source of motif is also indicated. Results of motif curation for GATA family TFs is displayed in *Figure 2—figure supplement 5*.

The following figure supplements are available for figure 2:

*Figure 2. continued on next page*

*Figure 2. Continued*

**Figure supplement 1**. *C. elegans* TFs adhere to established thresholds for motif inference.

**Figure supplement 2**. *C. elegans* TFs adhere to established thresholds for motif inference (continued).

**Figure supplement 3**. *C. elegans* TFs adhere to established thresholds for motif inference (continued).

**Figure supplement 4**. *C. elegans* TFs adhere to established thresholds for motif inference (continued).

**Figure supplement 5**. GATA TF motif clustering and identification of representative motifs.

available for the majority of the 292 TFs with motifs. We reasoned that the PBM data should facilitate direct comparison among TF sequence preferences, as they were generated using identical methodology. In addition, PBMs facilitate comparisons because they produce scores for individual DNA 8-mers. Thus, to complement the PWM analysis above, we examined as a composite 8-mer E-score data for all of the TFs analyzed in this study by using PBMs. *Figure 3* illustrates that the 8-mers recognized by each individual protein are in general distinct and further highlights the distinctiveness of the sequences preferred by different TFs that share the same type of DBD. For example, *C. elegans* homeodomain and Sox TFs display different sequence preferences that largely reflect the known subclasses (*Figure 3* and data not shown; all data and motifs are available in the Cis-BP database (see 'Data Access' section below)). We also observed subtle differences in Forkhead DNA-sequence preferences: despite the motifs having similar appearance, the proteins prefer slightly different sets of 8-mers, as previously observed using PBM data (*Badis et al., 2009*; *Nakagawa et al., 2013*). Other large *C. elegans* TF families display undocumented and unexpected diversity in their DNA-sequence preferences, which we next examined in greater detail.

## Complex relationships between protein sequences and motifs recognized by the NHR family

Previously, the literature contained motifs for only eight of the 271 *C. elegans* NHRs, while motifs for an additional 13 could be predicted from orthologs and paralogs (*Hochbaum et al., 2011*; *Weirauch et al., 2014*). It has also been reported that additional *C. elegans* NHRs bind sequences similar to those bound by their counterparts in other vertebrates (*Van Gilst et al., 2002*), but the data available do not lend itself to motif models that can be used for scanning. We obtained new PBM data for 20 *C. elegans* NHRs (*Figure 4*), among which only one had a previously known motif (DAF-12, which yielded a motif identical to one found by ChIP–chip [*Hochbaum et al., 2011*]). None of the remaining 19 could have been predicted by simple homology; due to their widespread divergence, and absence of motifs for most NHRs, few motifs can be predicted by homology among the *C. elegans* NHR class at our threshold for motif prediction (70% identity for NHRs). However, these 19 new NHR motifs do lead to predicted motifs for eight additional *C. elegans* NHRs.

The most striking feature of the NHR motifs is their diversity, but an equally surprising observation is that very different NHRs can bind very similar sets of sequences. Data from the 27 NHRs that have been analyzed by PBMs in our study or others are shown in *Figure 4*. We obtained 13 different groups of motifs, using the PWMclus methodology described above (indicated by shading of dendrogram labels in *Figure 4*). We expected that all 27 of these NHRs might have yielded a distinct motif, as no two are more than 70% identical to each other. In several cases, however, NHRs with very different overall DBD sequences (below the threshold for predicting motif identity) in fact display similar sequence preferences, while more similar NHR TFs often bind different motifs, as the shading on the labels in *Figure 4* does not strictly reflect the dendrogram. We also note that the data for individual 8-mers appear more complex than the motif groups capture (see heatmaps in *Figure 4*). For example, the individual 8-mer scores for TFs represented by the two largest groups of motifs—sets binding sequences related to G(A/T)CACA and (A/T)GATCA, respectively—indicate that they may in fact possess distinct DNA-sequence preferences (*Figure 4*, top and bottom). These subtle and complex differences are presumably obscured by the motif derivation process, which tends to produce degenerate (i.e., low

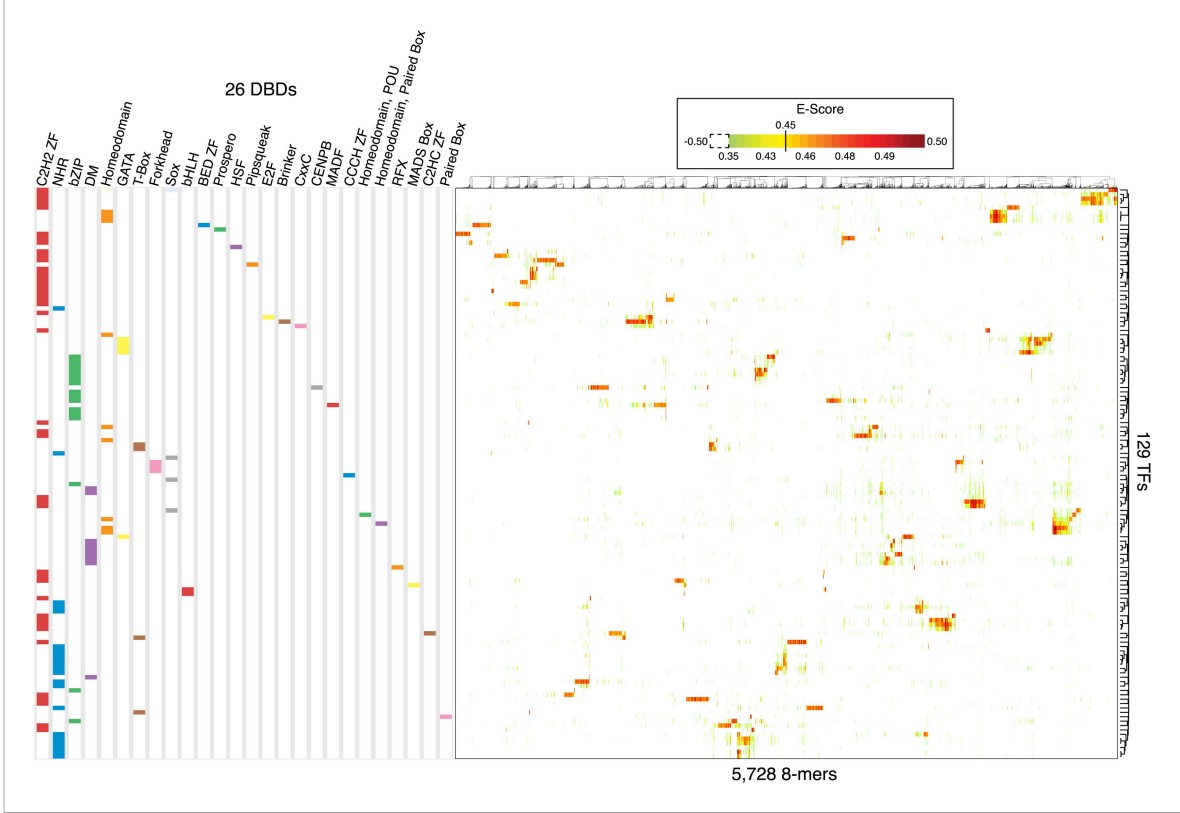

**Figure 3**. Overview of 8-mer sequences preferences for the 129 *C. elegans* TFs analyzed by PBM in this study. 2-D Hierarchical agglomerative clustering analysis of E-scores performed on all 5728 8-mers bound by at least one TF (average E > 0.45 between ME and HK replicate PBMs). Colored boxes represent DBD classes for each TF. Average E-score data is available in ***Figure 3—source data 1***.

The following source data is available for figure 3:

**Source data 1**. Table showing 8-mers bound by at least one TF with an average E-score ≥0.45 for all the 129 *C. elegans* TFs analyzed by PBMs in this study.

information content) motifs for most of these TFs. In addition, or possibly as a consequence, the default correlation threshold used by the PWMclus algorithm groups these TFs together.

To examine the determinants of NHR sequence preferences more closely, we considered NHR RH sequences (***Figure 4***, middle). Of the 95 unique RH sequences found in *C. elegans* NHRs, 15 are found in our data, including multiple representatives of most of the populous RHs (our data contain ten of the 75 with RA-AA; 3 of the 19 with NG-KT; 2 of the 10 with NG-KG; and one of the seven with AA–AA). It is believed that identity in the RH corresponds to identity in sequence preference (***Van Gilst et al., 2002***); surprisingly, however, we found that TFs with identical RH sequences can bind very different DNA sequences. For example, NHR-177 shares the RA-AA RH with nine other NHRs examined in our study, yet binds a completely different set of sequences (resembling CGAGA, unlike the CACA-containing motifs of the others). Conversely, NHRs with different RH sequences can have very similar DNA-sequence preferences. NHR-66 and NHR-70, for example, differ at two of the four variable residues in the RH (AA-SA vs RA-AA), and share only ~49% amino acid identity (and NHR-66 contains a three-residue insertion). Yet, they bind highly overlapping sets of 8-mers and produce motifs featuring CTACA. Thus, there is an imperfect correspondence between identity in the RH and identity in DNA-binding sequence preferences, suggesting that additional residues within (or flanking) the DBD contribute to the specificity of *C. elegans* NHR proteins. These observations also show that, when NHRs with very different overall DBD sequences bind similar motifs, it is typically not due to the two proteins sharing the same RH.

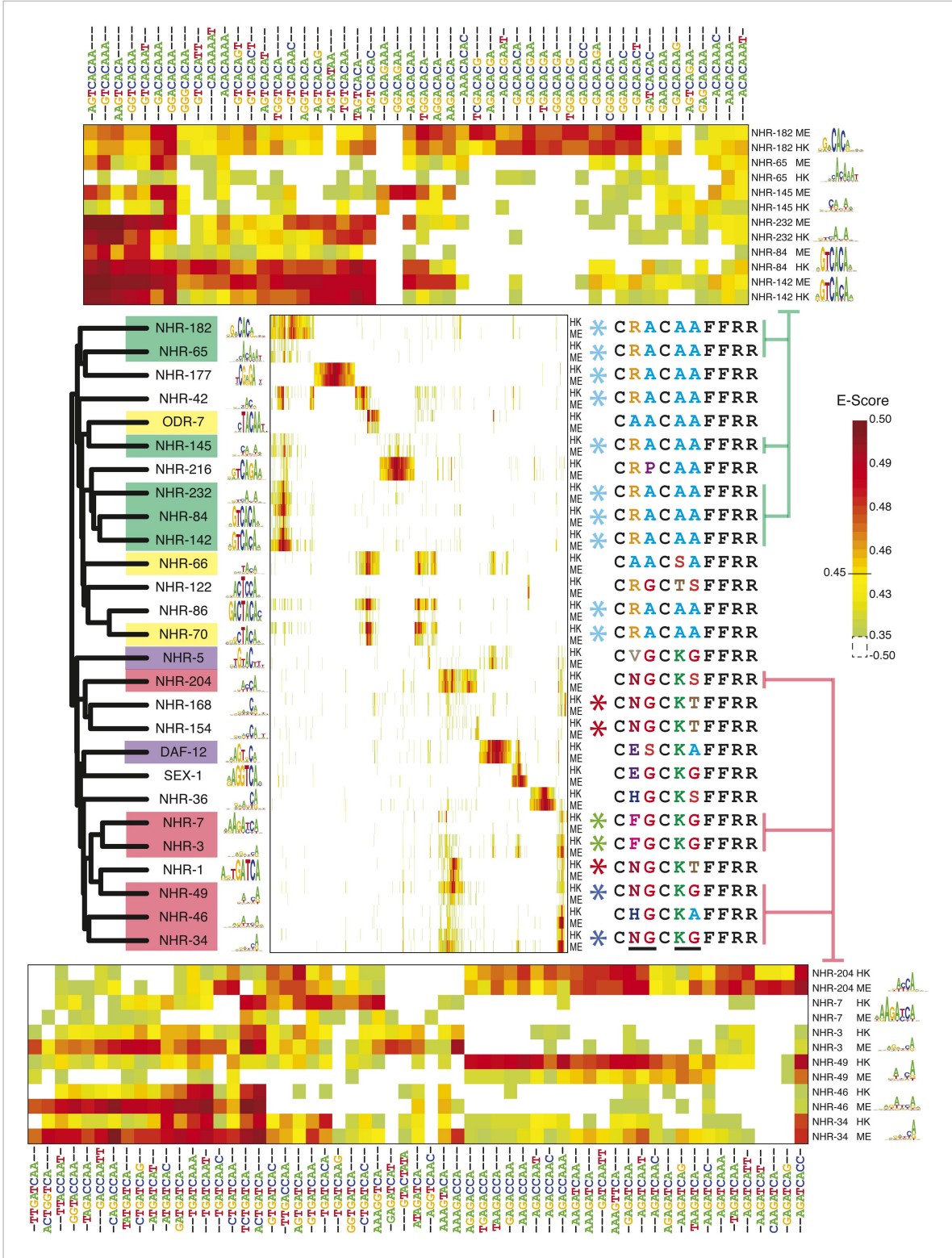

**Figure 4**. 8-mer binding profiles of NHR family reveal distinct sequence preferences. *Left*, ClustalW phylogram of nuclear hormone receptor (NHR) DBD amino acid sequences with corresponding motifs. TF labels are shaded according to motif similarity groups identified by PWMclus. *Center*, heatmap showing E-scores. NHRs are ordered according to the phylogram at left. The 1406 8-mers with E-score > 0.45 for at least one family member on at least one PBM array were ordered using hierarchical agglomerative clustering. Each TF has one row for each of two-replicate PBM experiments (ME or HK array designs). *Right*; recognition helix (RH) sequences for the corresponding proteins, with identical RH sequence types highlighted by colored asterisks.
*Figure 4. continued on next page*

Figure 4. Continued

Variant RH residues are underlined at bottom. *Right*, matrix indicates cluster membership according to PWMclus. *Top and bottom*, pullouts show re-clustered data including only the union of the top ten most highly scoring 8-mers (taking the average E-score from the ME and HK arrays) for each of the selected proteins. E-scores for k-mers in all three heatmaps are available in *Figure 4—source data 1*.

The following source data is available for figure 4:

**Source data 1**. Table showing 8-mer E-score profiles of NHRs analyzed by PBMs.

Only one NHR, SEX-1, produced a motif strongly resembling the canonical steroid hormone response element (SHRE) (GGTCA); SEX-1 shares three of four variable residues in the RH with canonical SHRE binding TFs such as the estrogen receptor (SEX-1: EG-KG; ER: EG-KA). Moreover, none of the NHRs examined produced a motif matching that of HNF4, the presumed ancestor of most *C. elegans* NHRs.

## Motifs for *C. elegans* C2H2 TFs are supported by the recognition code

We obtained new PBM data for 42 C2H2 ZF TFs (*Figure 5*), only one of which was previously known (*Figure 1—figure supplement 1*). Previously there were only six experimentally-determined *C. elegans* C2H2 motifs in the literature, and 11 that could be predicted by homology, all of which are well conserved in distant metazoans (members of KLF, SP1, EGR, SNAIL, OSR, SQZ, and FEZF families); seven of these are among our data and have PBM motifs consistent with those predicted (data not shown). Only two additional TFs (ZTF-25 and ZTF-30) can be assigned motifs by homology using our new data. Together, the new data and predictions bring the total number of *C. elegans* C2H2 TFs with motifs to 53 (~50% of the 107 C2H2s in our list of 763 TFs).

The C2H2 motifs are diverse (*Figure 5*), but unlike the NHR family, the molecular determinants of C2H2 DNA sequence specificities are more readily understood. The motifs we obtained are broadly consistent with previously determined relationships between DNA contacting residues and preferred bases (the so-called 'recognition code') (*Wolfe et al., 2000*), although the motifs predicted by the recognition code are not sufficiently accurate to be used in motif scans (median $R^2 = 0.21$ vs predictions made by an updated recognition code that surpasses all previous recognition codes when compared against gold standards (*Najafabadi et al., 2015*)). While most of the motifs are similar to those predicted by the recognition code (*Figure 5—figure supplement 1*), lower similarity is observed for TFs with unusual inter-C2H2 linker lengths and atypical zinc-coordinating residues (*Figure 5—figure supplement 1*). In some cases, differences in the motifs obtained from related C2H2 TFs can be rationalized: *Figure 5* (right) shows the example of paralogs EGRH-1 and EGRH-3, in which the motifs obtained by PBM closely reflect those predicted by the recognition code, which differ at several positions. *Figure 5* also shows the example of Snail homologs CES-1 and K02D7.2, in which a short linker between fingers 2 and 3 may explain the truncated motif in K02D7.2, and may also explain the differences previously observed between these two proteins in Y1H assays (*Reece-Hoyes et al., 2009*).

## Unexpected diversity in T-box DNA-binding specificities

We obtained motifs for four nematode-specific T-box TFs (i.e., lacking one-to-one orthologs in other phyla): TBX-33, TBX-38, TBX-39, and TBX-43. In addition, TBX-40 was previously analyzed by PBM, and our motif for the related protein TBX-39 (93% identical) is very similar. T-box TFs can bind to dimeric sites, with the characteristic spacing and orientation varying among different T-box proteins (*Jolma et al., 2013*). The monomeric sequence preference (resembling 'GGTGTG') is thought to be constant, however, as it is observed across different T-box classes and in distant phyla (*Sebé-Pedrós et al., 2013*; *Weirauch et al., 2014*). Strikingly, our new PBM data indicate that monomeric T-box sites can also vary considerably (*Figure 6A*). While the motifs for TBX-38 and TBX-43 are highly similar to the canonical 'GGTGTG' motif, TBX-33, TBX-39, and TBX-40 exhibit novel recognition motifs.

The primary determinants of sequence specificity of T-box TFs are believed to reside in amino-acid residues located in α-helix 3 and the $3_{10}$-helixC, which contact the major and minor groove, respectively (*Muller and Herrmann, 1997*; *Coll et al., 2002*; *Stirnimann et al., 2002*), and indeed, the DNA contacting residues in TBX-33, TBX-39, and TBX-40 are different from those in T-box TFs that bind the canonical motif (*Figure 6—figure supplement 1*). In addition, TBX-39 and TBX-40 exhibit

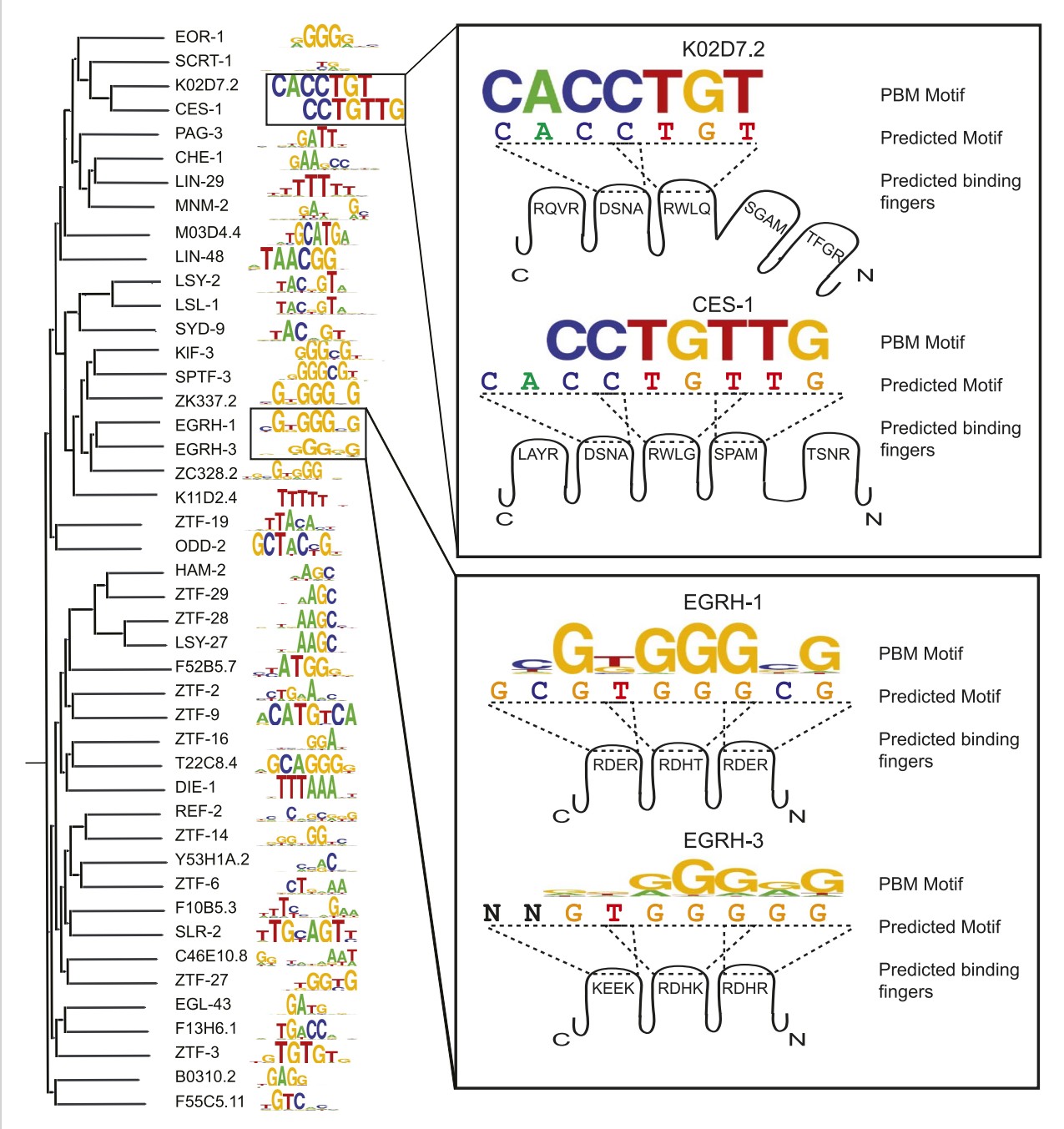

**Figure 5**. C2H2 motifs relate to DBD similarity and to the recognition code. *Left*, ClustalW phylogram of C2H2 zinc finger (ZF) amino acid sequences with corresponding motifs. *Right*, examples in which motifs predicted by the ZF recognition code are compared to changes in DNA sequences preferred by paralogous C2H2 ZF TFs. Cartoon shows individual C2H2 ZFs and their specificity residues. Dashed lines correspond to 4-base subsites predicted from the recognition code.

The following figure supplement is available for figure 5:

**Figure supplement 1**. Comparison of C2H2 ZF recognition model with motifs derived PBM.

sequence deletion in the 'variable region', and TBX-33 has an 18 amino acid insertion in the region leading up to the β-strand e′, which could also potentially alter sequence preferences via structural rearrangements.

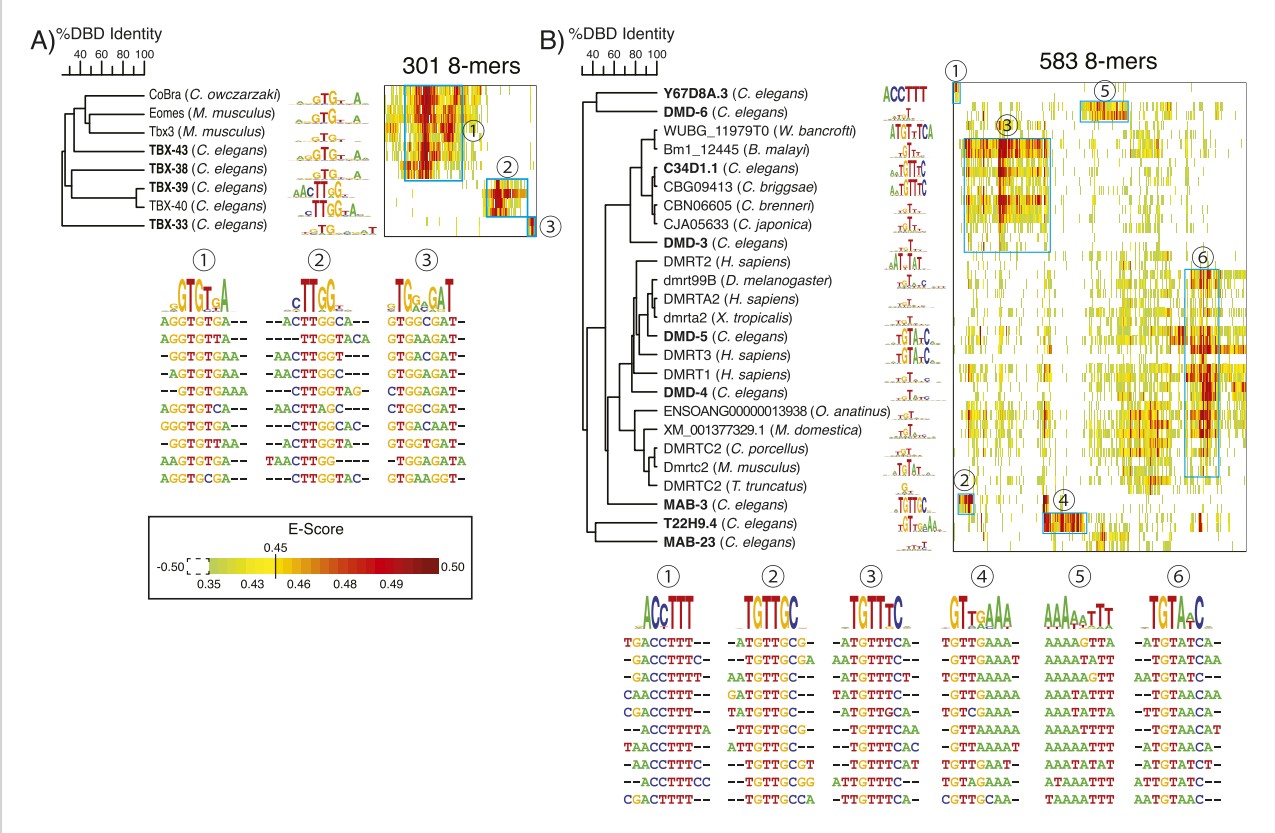

**Figure 6**. Nematode-specific sequence preferences in T-box and DM TFs. PBM data heatmaps of preferred 8-mers for T-box (**A**), and DM (**B**) TFs. TFs are clustered using ClustalW; 8-mers were selected (at least one instance of E > 0.45) and clustered using hierarchical agglomerative clustering, as in **Figures 4, 5**. Ten representative 8-mers (those with highest E-scores) are shown below for each of the clusters indicated in cyan. *C. elegans* TFs with data from this study are bolded. E-score data for T-box and DM TFs available in **Figure 6—source data 1**.

The following source data and figure supplement are available for figure 6:

**Source data 1**. Table showing 8-mer E-score profiles of T-box and DM TFs from *C. elegans* and other metazoans that have been analyzed by PBMs.

**Figure supplement 1**. T-box sequence alignments and the crystal structure of mTBX3 illustrate *C. elegans* specific variations.

## Variation in motifs for DM domains highlights nematode-specific expansions

DM TFs are well studied because of their established roles in sex determination, and previous analyses established that different DM TFs often bind distinct motifs that typically contain a TGTAT core, including *Drosophila* doublesex, for which the family is named (**Gamble and Zarkower, 2012**). *C. elegans* and other nematodes encode several lineage-specific DM TFs in addition to orthologs shared across metazoans, with eight of the eleven *C. elegans* DM domains having less than 85% identity (our threshold for DM motif prediction) to any DM domain in insects and vertebrates (**Weirauch et al., 2014**). Accordingly, most of the *C. elegans* DM domains have highest preference for sequences that are different from TGTAT, although in all but two cases the motifs do contain a TGT (**Figure 6B**). DM domains encode intertwined CCHC and HCCC-zinc binding sites and are hypothesized to bind primarily in the minor groove (**Zhu et al., 2000**; **Narendra et al., 2002**). A DNA-protein structure has not yet been described for any DM protein, however; mapping the determinants of their variable DNA sequence preferences will therefore require further study.

## Motif enrichment in Y1H and ChIP-seq data

We next examined whether motifs from our collection correspond to modENCODE TF ChIP-seq data (**Araya et al., 2014**), and to TF prey—promoter bait interactions from Y1H experiments

([*Reece-Hoyes et al., 2013*] and JF-B and AJMW, unpublished data). Among the 40 TFs analyzed by ChIP-seq and present in our motif collection, peaks for 20 TFs displayed central enrichment of motif scores (*q*-value < 0.05) using the CentriMo algorithm on the top 250 peaks (*Bailey and Machanick, 2012*) (*Figure 7*). Similarly, among 145 TFs both analyzed by Y1H and present in our motif collection, motif affinity scores for 103 were significantly enriched (Mann–Whitney U test; *q*-value < 0.05) among promoter sequences scoring as positive by Y1H, relative to those scoring as negative by Y1H (*Figure 7—figure supplement 1*). The correspondence among these data sets is presumably imperfect due to indirect DNA binding in vivo, and/or the impact of chromatin and cofactors on binding site selection (*Liu et al., 2006*), both of which occur in *C. elegans* and yeast. We note that, among the 25 TFs that are present in Y1H data, ChIP-seq data, and our motif collection, 11 are only significantly enriched in Y1H (using the cutoff above), five are only significantly enriched in ChIP-seq, and only five are significantly enriched in both. Thus, most motifs (21/25; 84%) can be supported by independent assays, although the in vivo assays appear to capture different aspects of TF binding. Overall, the clear relationship between our motifs and independent data sets strongly supports direct in vivo relevance of the motifs.

We also examined whether we could detect multimeric or composite motifs (CMs) in existing ChIP-seq data sets by searching for the enrichment of patterns in which there is fixed spacing and

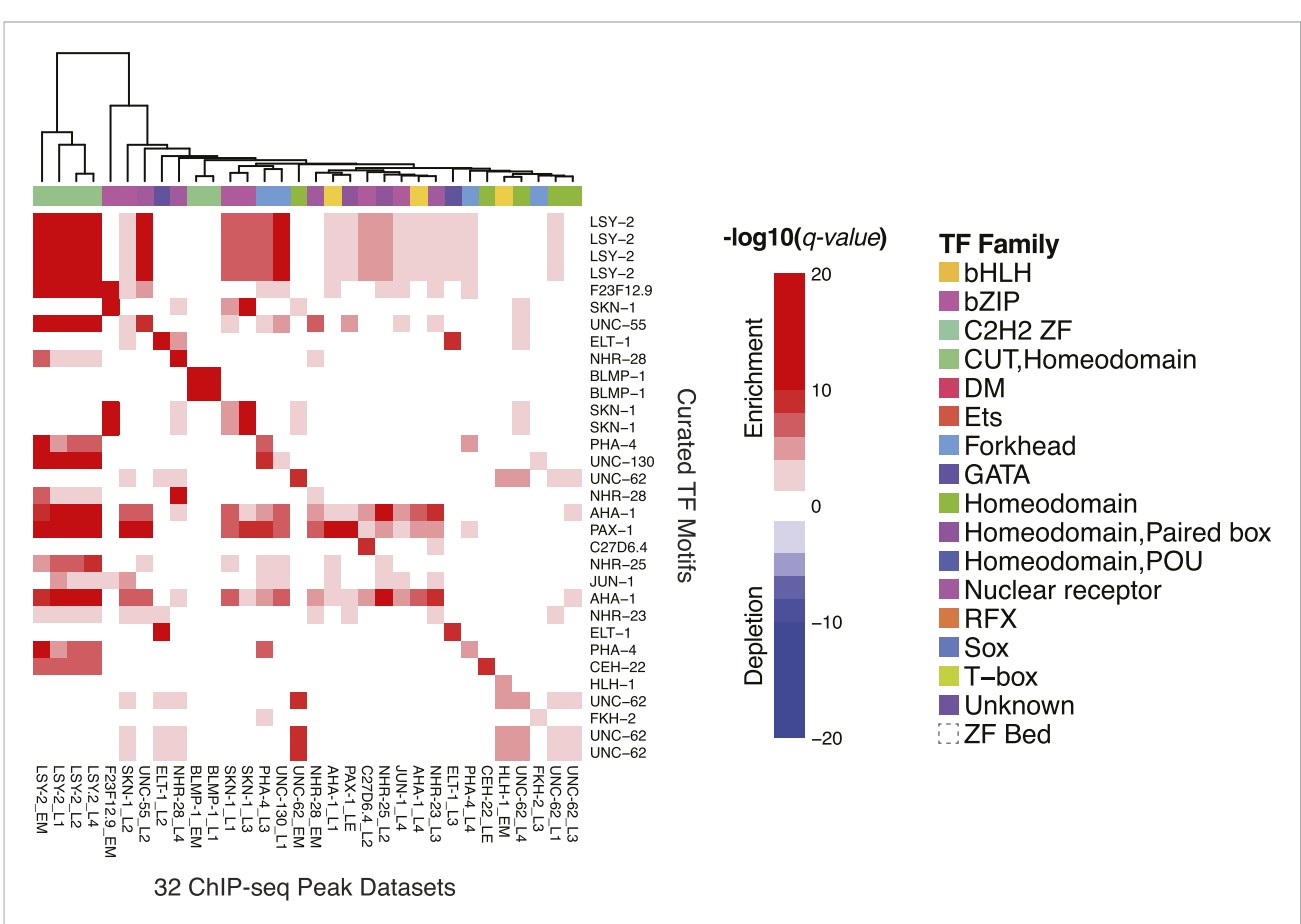

**Figure 7**. The *C. elegans* curated motif collection explains ChIP-seq and Y1H TF binding data. Heatmap of CentriMo −log10 (*q-values*) for central enrichment of TF motifs in the top 250 peaks for each ChIP experiment. Motif enrichment in Y1H data is presented in *Figure 7—figure supplement 1*. Both ChIP-seq and Y1H heatmaps use a common colour and annotation scale. Heatmaps are symmetric with duplicate rows to ensure the diagonal represents TF-motif enrichment in it's matching data set(s). Red and blue coloring depicts statistically significant enrichments and depletions (*q* ≤ 0.05).

The following figure supplement is available for figure 7:

**Figure supplement 1**. The *C. elegans* curated motif collection explains Y1H TF binding data.

orientation between two or more motifs within the peaks, one of which corresponds to the TF that was ChIPed. We identified 185 significantly enriched CMs (see 'Materials and methods') involving 11/40 ChIPed TFs, and 14 different TF families (including partner motifs) (*Figure 8*, *Figure 8—source data 1*). As an example, the most highly significant result involves NHR-28 in which the six-base core sequence 'ACTACA' (which could correspond to NHR-28 or NHR-70) is found repeated in both dimeric and trimeric patterns (*Figure 8A*, top). We also identified a CM involving LSY-2 (a C2H2 ZF protein) and NHR-232 with a spacing of one base between the core motifs (*Figure 8A*, middle). A subset of these instances included a ZIP-6 (bZIP) motif at a one base distance 3′ of the NHR-232 motif, yielding a multi-family trimeric CM (*Figure 8A*, bottom).

PWMclus grouped the 185 CMs into 37 clusters (*Figure 8—figure supplements 1–5*). Most of the CMs were identified repeatedly for the same TF ChIPped in different developmental stages; these instances were considered separately in the analysis above and highlight the robustness of the observations. Some of the clusters also correspond to CMs containing motifs for related TFs, demonstrating robustness to the exact motif employed. Our methodology allowed the individual motifs to overlap, and half of the 37 CM clusters represent such overlaps. However, the majority of overlaps occur in the flanking low-information-content sections of motifs, such that most the 37 CM clusters resemble a concatenation of two motif 'cores' with or without a small gap (1–4 bases). Surprisingly, 17 of the 37 clusters (∼46%) were obtained from the embryonic ChIP-seq data for the poorly-characterized, essential bZIP protein F23F12.9 (ZIP-8), which is most similar to human ATF TFs and binds both the ATF site and the CREB site (Figure 8—figure supplements 1–3). In total, these results suggest that multimeric interactions within and between TF families may be a prevalent phenomenon in *C. elegans*.

## Motif enrichment in tissue and developmental-stage specific expression data

To identify potential roles for TFs in the regulation of specific groups of functionally related genes, we asked whether the set of promoters containing a strong motif match to each TF (FIMO p-value < $10^{-4}$ in the region −500 to +100 relative to TSS) overlapped significantly with any tissue expression (*Spencer et al., 2010*), Gene Ontology (GO) categories (*Ashburner et al., 2000*), or Kyoto Encyclopedia of Genes and Genomes (KEGG) pathways (*Kanehisa et al., 2014*) (Fisher's exact test, one-sided probability, false discovery rate (FDR) corrected $q$-value < 0.05). We obtained dozens of significant relationships (*Figure 9*), including known roles for GATA TFs in the regulation of intestinal gene expression (and related GO categories) (*Pauli et al., 2006*; *McGhee, 2007*), HLH-1 in the regulation of muscle gene expression (*Fukushige et al., 2006*), DAF-19 (an RFX TF) in the regulation of ciliary genes (*Swoboda et al., 2000*), and PHA-4 in development of the pharynx (*Gaudet and Mango, 2002*) (boxed in *Figure 9*). We also note that the association of the motif for ZTF-19 (PAT-9), a C2H2 ZF protein, with genes expressed in L2 body wall muscle tissue is consistent with observed expression patterns for this gene in body wall muscle, as well as defective muscle development in a mutant (*Liu et al., 2012*). The ZTF-19 binding motif may, therefore, enable the identification of specific downstream targets. Most of the associations in *Figure 9*, however, appear to represent putative regulatory interactions, suggesting that the motif collection can be used to gain new biological insight.

## Discussion

The collection of motifs described here will further advance *C. elegans* as a major model system for the study of gene regulation. TF DNA-binding motifs enable dissection of promoters, prediction of new targets of TFs, and identification of putative new regulatory mechanisms. Statistical associations between motif matches in promoters and expression patterns or functional categories of genes also provide a ready starting point for directed experimentation, for example, analysis of gene expression in mutants. Apparent position and orientation constraints between motif matches also suggest functional relationships. Our observation that the largely unstudied bZIP TF F23F12.9 (ZIP-8) was involved in almost half of all CMs identified in this study suggests that it may function as a cofactor for targeting to open chromatin: pioneer TF activity and partnering with other TFs has previously been proposed for other Creb/ATF proteins in mouse embryonic stem cells (*Sherwood et al., 2014*).

A key observation in this study is that all the large groups of TFs in *C. elegans* are malleable in their DNA-binding sequence preferences. The NHR is a striking case, even more so when we consider that

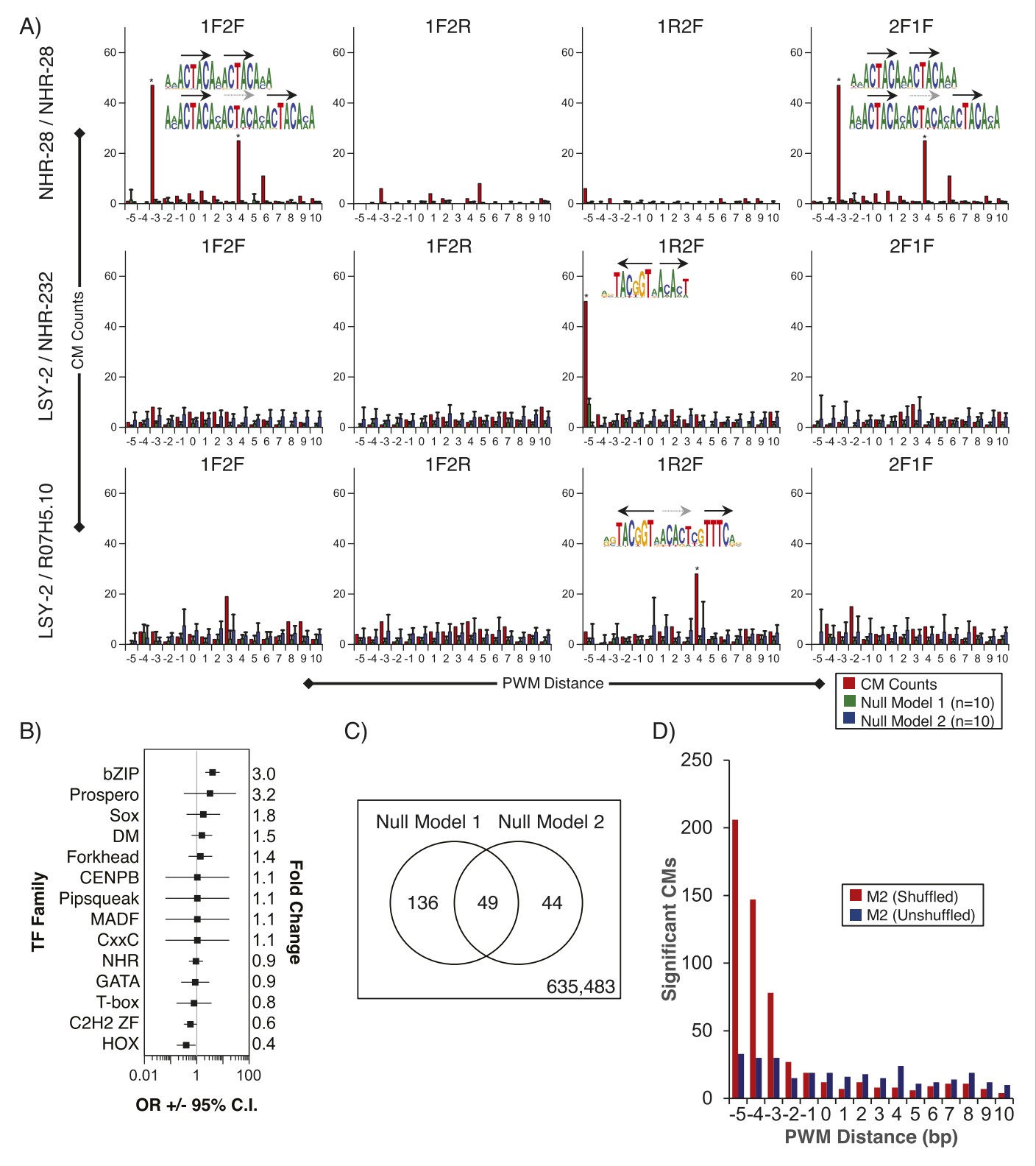

**Figure 8**. Composite motifs enriched in *C. elegans* ChIP-seq peaks. (**A**) Stereospecificity plots showing enriched CM configurations for pairs of TF motifs. The identical '1F2F' and '2F1F' results in A (top row) demonstrate homodimer and homotrimer CMs, while those involving LSY-2, NHR-232, and R07H5.10 demonstrate heterodimer and heterotrimer CMs (middle and bottom rows, respectively). Black arrows represent orientation of the motif within CMs, while gray dashed arrows designate shadow motifs within trimeric CMs. Error bars are ± S.D., *corrected p < 0.05. (**B**) Forest plot of odds ratios for TF family

*Figure 8. Continued*

enrichment in CMs vs input TF list. (**C**) Venn diagram showing overlap of significant CMs identified by null model 1 (dinucleotide shuffled sequence) and null model 2 (motif shuffling). (**D**) Number of significant CMs identified relative to dinucleotide scrambled sequences using shuffled and non-shuffled non-ChIPed motifs, as a function of motif pair distance.

The following source data and figure supplements are available for figure 8:

**Source data 1**. Table displaying enrichment statistics, spacing and orientations between PWMs for CMs identified in modENCODE ChIP-seq data.

**Figure supplement 1**. Summary of clustered CMs enriched in *C. elegans* ChIP-seq peaks.

**Figure supplement 2**. Summary of clustered CMs enriched in *C. elegans* ChIP-seq peaks (continued).

**Figure supplement 3**. Summary of clustered CMs enriched in *C. elegans* ChIP-seq peaks (continued).

**Figure supplement 4**. Summary of clustered CMs enriched in *C. elegans* ChIP-seq peaks (continued).

**Figure supplement 5**. Summary of clustered CMs enriched in *C. elegans* ChIP-seq peaks (continued).

our motifs encompass only monomeric binding sites. A previous analysis classified the *C. elegans* NHRs into four subtypes, on the basis of their RH sequences, and predicted that all of those in Class I (those most similar to canonical NHRs such as the estrogen receptor) would likely bind canonical SHRE GGTCA subsites (*Van Gilst et al., 2002*). Instead, we find that those in Class I (the entire lower half in *Figure 4*) bind a wide range of sequences, and that the RH cannot be the only determinant of sequence specificity. Our observations are consistent with the previous demonstration that mutation of one or a few residues in the NHR RH can result in dramatic changes in sequence preferences, but that mutations elsewhere in the DBD play a role in sequence selectivity (e.g., *McKeown et al., 2014*). Recent analyses of other DBD classes (e.g., C2H2 and Forkhead) also highlight the importance of residues beside the canonical specificity residues (*Nakagawa et al., 2013*; *Siggers et al., 2014*). Together, these analyses strongly confirm that alteration of binding motifs is widespread among TF classes throughout evolution.

Our study experimentally determined motifs for 129 TFs, all but five of which were previously unstudied, bringing the total number of *C. elegans* TFs with motifs to 292 (including predicted motifs and data already in the literature). We estimate that the remaining 453 *C. elegans* TFs encode as many as 409 different DNA-binding motifs, most of which correspond to NHRs, C2H2 ZFs, bHLHs, and homeodomains (*Supplementary file 3*). Additional effort will thus be required to obtain a complete motif collection. For instance, even with our new motifs, and including motifs predicted by homology, coverage for the *C. elegans* NHR family is only 17%.

For some classes of DBDs, most of the PBM assays yielded negative data. The NHRs in particular yielded only 20% success (27/135). In addition, of the 108 that failed by PBM, we have tested 100 by Y1H, of which only 17 succeeded (three or more detected interactions; data not shown). We observed no obvious property of their DNA-contacting residues that strongly predicts success or failure and hypothesize that requirement for ligand binding, dimerization, cofactors, or protein modifications may represent other potential explanations for failures in heterologous assays. Like human NHRs, the *C. elegans* NHRs have a ligand-binding domain that is distinct from the DNA-binding domain and is thought to primarily regulate interactions with coactivators and corepressors (*Sonoda et al., 2008*). Thus, ligand-dependent DNA binding seems unlikely, especially for an in vitro assay. Yeast two-hybrid screens have identified several interactions between different NHRs (*Simonis et al., 2009*; *Reece-Hoyes et al., 2013*), suggesting that heterodimerization may be prevalent. If DNA binding is often dependent on heterodimerization, then ChIP-seq should often succeed where PBMs and Y1H fail, and heterodimeric motifs should be identified. To the best of our knowledge, however, there is only one published ChIP-seq data set for a *C. elegans* NHR that has yielded a motif de novo (NHR-25) (*Araya et al., 2014*; *Boyle et al., 2014*). We did not test NHR-25 by PBM, although our predicted monomeric motif (from *Drosophila* Ftz-f1) resembles the motif identified by ChIP-seq. As noted above, our motif for DAF-12 is also consistent with the motif obtained by ChIP–chip (*Hochbaum et al., 2011*). In

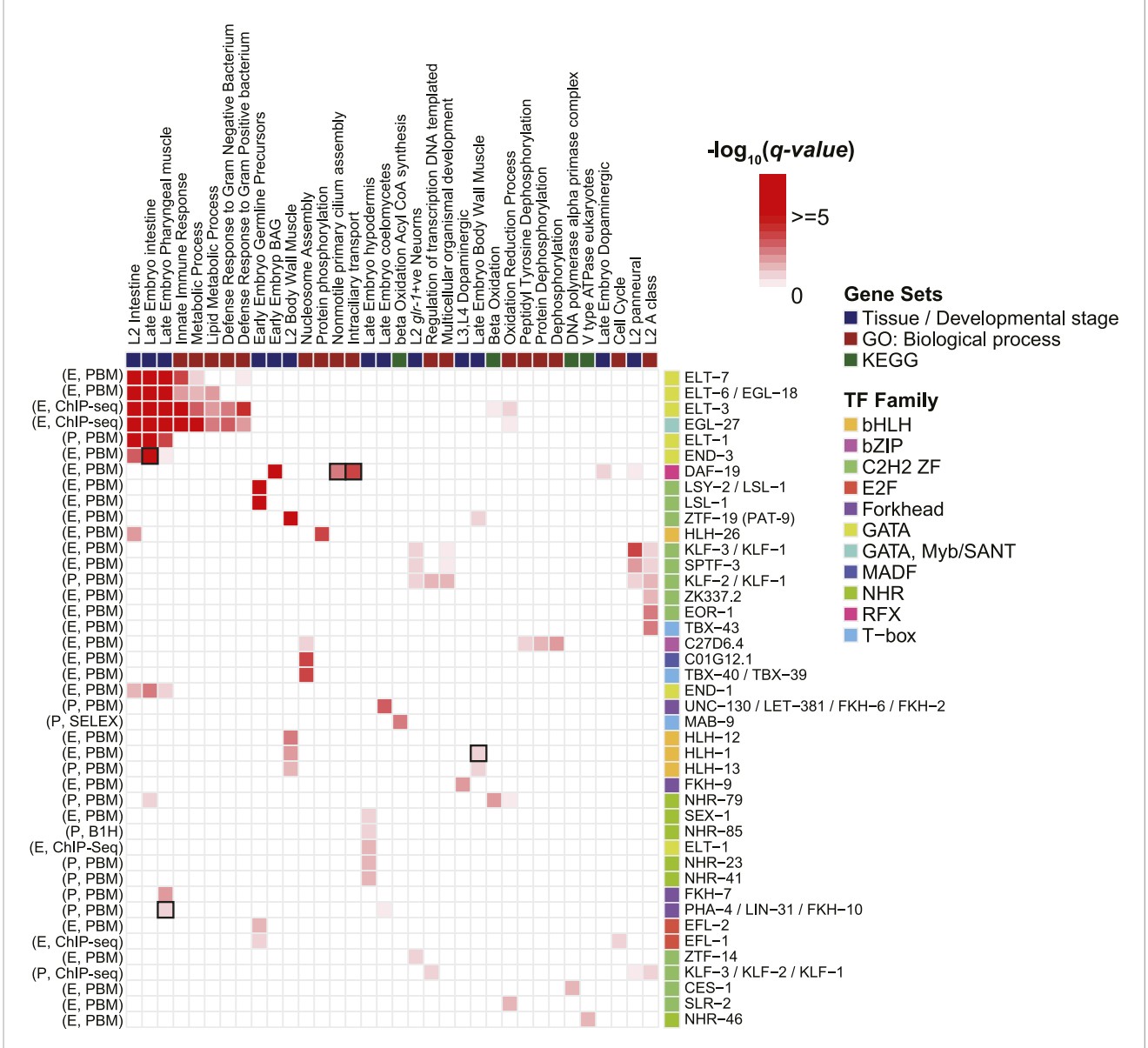

**Figure 9**. Enrichment of motifs upstream of gene sets. Each row of the heatmap represents a motif from our curated collection that is enriched ($q < 0.05$) in at least one gene set category. Known regulatory interactions between TFs and gene sets are highlighted (black outlines). 'E' indicates experimentally determined motifs; 'P' indicates predicted motifs. Source of motif is also indicated. Enrichment $q$-values are available in *Figure 9—source data 1*.

The following source data is available for figure 9:

**Source data 1**. Table of motif enrichments $-\log_{10}$ (p-values) in the promoters of gene set categories identified from KEGG pathway modules, Gene Ontology processes, and tissue/developmental stage specific expression lists.

support of heteromeric binding, however, CMs involving NHRs and other TF families were prevalent in modENCODE ChIP-seq data (*Figure 8—source data 1*). Further analysis will be required to explore the role of multimerization in *C. elegans* TF DNA binding and gene regulation.

Finally, we note that there are numerous similarities between the TF collections of human, *Drosophila*, and *C. elegans*. First, the total number TFs containing a canonical DBD varies only by a factor of ~2 (~1734, 701, and 744 for human, *Drosophila*, and *C. elegans*, respectively [*Weirauch et al., 2014*]). The total number of groups of closely related paralogs (taken using the thresholds

established in *Weirauch et al. (2014)*), which should approximate the number of distinct motifs that can eventually be expected, also varies by only a factor of ~2 (the TFs fall into 1339, 656, and 632 groups of proteins expected to have very similar sequence specificity, respectively). Our study also brings the proportion of *C. elegans* TFs with a known or predicted motif much closer to the proportion in human and *Drosophila*, (56.1%, 54.1%, and 39.2% for human, *Drosophila*, and *C. elegans*, respectively). In addition, all three species possess a large number of diverse lineage-specific TFs, which are known or expected to bind different motifs: in human, ~700 C2H2 ZF TFs; in *Drosophila*, 230 C2H2 ZF TFs; and in *C. elegans*, 266 NHRs, 99 C2H2 ZF TFs and—as we show here—roughly a dozen T-box and DM TFs. Thus, despite widespread conservation in TF number and gene expression (*Stuart et al., 2003*) among metazoans, extensive rewiring of the metazoan *trans* regulatory network is apparently common.

## Materials and methods

### Selection of TFs for analysis

We compiled a list of 874 known and putative *C. elegans* TFs. We took 740 from build 0.90 of the Cis-BP database (*Weirauch et al., 2014*), plus additional candidate TFs that lack canonical DBDs (*Reece-Hoyes et al., 2005*). We considered selecting each TF for characterization using the following criteria:

1. Include the TF if its characterization would provide multiple-motif predictions for other TFs based on established prediction thresholds for the given DBD class (*Weirauch et al., 2014*);
2. Include the TF if it is a member of a DBD class with relatively little available motif information;
3. Include the TF if it has a known important biological role;
4. Exclude the TF if it has already been characterized by PBM in another study;
5. Exclude the TF if it is a member of a DBD class with a low-PBM success rate;
6. Exclude the TF if the resulting construct would be excessively long (for example, exclude C2H2 ZF TFs with many DBDs).

### Cloning of *C. elegans* TF DBDs

We identified putative DBDs for all TFs by scanning their protein sequences using the HMMER tool (*Eddy, 2009*), and a collection of 81 Pfam (*Finn et al., 2010*) models taken from (*Weirauch and Hughes, 2011*), as described previously (*Weirauch et al., 2014*). For some TFs, we could not identify DBDs using this procedure. In such cases, DBDs were manually detected by lowering HMMER scanning thresholds, using DBDs annotated in the SMART database (*Letunic et al., 2012*), or performing literature searches. Using the above criteria for selection of TFs, we initially chose 398 TFs from the Walhout clone collection for characterization. We designed primers (*Supplementary file 4*) to clone open reading frames (ORFs) comprising the DBDs plus additional flanking sequences (50 endogenous amino acid flanking residues, or until the end of the protein). We inserted the resulting sequences using *Asc*I and *Sbf*I restriction sites into a modified T7-driven expression vector (pTH6838) that expresses N-terminal GST fusion proteins (*Supplementary file 5*). In the first round of cloning, we attempted cloning using both individual plasmids and pooled mRNA (by RT-PCR) or cDNA. After PBM analysis with the resulting clones, we then considered remaining uncharacterized TFs and selected an additional 154 TFs using the same criteria as above. The DBDs and flanking bases of these TFs were created using gene synthesis (BioBasic, Canada) and inserted into vectors as described above. Primers and insert sequences are provided on our project Web site. All clones were sequence verified.

### PBMs and data processing

PBM laboratory methods were identical to those described previously (*Lam et al., 2011*; *Weirauch et al., 2013*). Each plasmid was analyzed in duplicate on two different arrays with differing probe sequences. Microarray data were processed by removing spots flagged as 'bad' or 'suspect' and employing spatial de-trending (using a $7 \times 7$ window centered on each spot) (*Weirauch et al., 2013*). Calculation of 8-mer Z- and E-scores was performed as previously described (*Berger et al., 2006*). Z-scores are derived by taking the average spot intensity for each probe containing the 8-mer, then subtracting the median value for each 8-mer, and dividing by the standard deviation, thus yielding a distribution with a median of zero and a standard deviation of one. E-scores are a modified version of the AUROC statistic, which consider the relative ranking of probes containing a given 8-mer, and

range from −0.5 to +0.5, with E > 0.45 taken as highly statistically significant (*Berger et al., 2008*). We deemed experiments successful if at least one 8-mer had an E-score > 0.45 on both arrays, the complimentary arrays produced highly correlated E- and Z-scores, and the complimentary arrays yielded similar PWMs based on the PWM_align algorithm (*Weirauch et al., 2013*).

## Generation of PWMs from PBMs

Motif derivation followed steps as outlined previously (*Weirauch et al., 2014*). Briefly, to obtain a single representative motif for each protein, we generated motifs for each array using four different algorithms: BEEML-PBM (*Zhao and Stormo, 2011*), FeatureREDUCE (manuscript in prep, source code available at http://rileylab.bio.umb.edu/content/software), PWM_align (*Weirauch et al., 2013*), and PWM_align_Z (*Ray et al., 2013*). We scored each motif on the complimentary array using the energy scoring system utilized by the BEEML-PBM algorithm (*Zhao and Stormo, 2011*). We then compared these PWM-based probe score predictions with the actual probe intensities using (1) the Pearson correlation coefficient and (2) the AUROC of 'bright probes' (defined by transforming all probe intensities to Z-scores, and selecting probes with Z-scores $\geq$ 4), following (*Weirauch et al., 2013*). Finally, we chose a single PWM for each DBD construct using these two criteria, as previously described (*Weirauch et al., 2014*).

## Expert curation

For every TF with motif information, we selected a representative motif, or motifs if that TF appears to have multiple-binding modes (e.g., multimers), using the following scheme. If the TF has an experimentally derived motif, it is selected as the primary motif. If there are multiple such motifs, we selected one that was derived in vitro, if any. If the TF had multiple in vitro motifs, then we ranked PBM > B1H > SELEX, to maximize comparability among motifs, and excluded motifs that are inconsistent with known motifs for the same or highly related proteins. If the TF had only predicted motifs, we selected a motif from a highly similar TF that is: preferably derived from an in vitro method (PBM > B1H > SELEX); assigned to the cluster that contains the majority of motifs for that TF in our PWMclus analysis; consistent with known DBD preferences; and best supported by ChIP-seq or Y1H data, if available.

## Motif enrichment with Y1H and ChIP-seq data

We calculated motif enrichment in ChIP-seq peaks using CentriMo (*Bailey and Machanick, 2012*), which uses TF motifs to look for central enrichment of motifs in ChIP-seq peaks, as an indication of direct binding by that TF. We obtained ChIP-seq peaks from the *C. elegans* modENCODE consortium (*Araya et al., 2014*). We used the top 250 peaks ranked by Irreproducible Discovery Rate (*Landt et al., 2012*) as the input data sets. We scored the curated set of motifs for TFs with peak data sets across all the peaks. We report false discovery rate (FDR)-adjusted *p*-values for a motif's central enrichment in TF-peak data sets.

For yeast one-hybrid (Y1H) data, we assigned motif scores to promoter bait sequences using the BEEML scoring system (*Zhao et al., 2009*). We included TFs in the analysis only if they bound five or more promoters in Y1H (those with 3 or 4 promoters bound were excluded to minimize sampling error in Mann–Whitney tests). We scored only the promoter-proximal 500 bp of Y1H bait sequences, as activating TF-binding sites are mainly effective within a few hundred bases of TSS in *Saccharomyces cerevisiae* (*Dobi and Winston, 2007*). We calculated motif enrichment or depletion for motifs using a two-tailed Mann–Whitney *U* test and reported with FDR-corrected *p*-values, with Y1H interactors as positives and the remaining non-interacting baits as the background.

We performed composite motif (CM) analysis by scanning 77 *C. elegans* ChIP-seq top 250 peak sequences for all pairwise combinations between the 40 ChIPed TFs (using the curated list of PWMs) and 129 PBM-derived PWMs from this study. Relative PWM spacing was restricted to −5 (overlapping) to +10 (gapped) bp separation, with four possible stereospecific arrangements of TFs: TF-1 forward TF-2 forward (1F2F), TF-1 forward TF-2 reverse (1F2R), 2R1F, and 2F1F, yielding 64 stereospecific combinations. We identified sequence matches using the standard log-likelihood scoring framework (*Stormo, 2000*), with a threshold of 0.50*max_score for each PWM, where max_score is the highest possible score for the given PWM. We created 10 sets of background sequences by scrambling the input sequences (maintaining dinucleotide frequencies). We calculated sample Z-scores and *p*-values by comparing the number of sequence matches observed in the 'real' sequence to the number

observed in the random sequences and applied a Bonferroni correction to each *p*-value. To identify significant composite motifs, we filtered to retain only results with sequence match counts ≥10% of the number of input peak sequences and Bonferroni-adjusted p-values ≤ 0.05 (alpha = 0.05). We also considered an alternative null model, in which we shuffled the non-ChIPed motif, and counted matches in the original DNA sequences (this procedure was repeated 10×). Overall, we found very good agreement using this approach and our original null model. Out of the 635,712 possible patterns we tested, both methods call 635,483 insignificant, both call 49 positive, and they disagree on 180 (*Figure 8C*). *Figure 8D* plots the number of significant hits identified relative to dinucleotide scrambled sequences using shuffled (blue) and non-shuffled (red) non-ChIPed motifs. This plot indicates, however, that the shuffled motif null model over-estimates the significance of CMs as the overlap of their constituent motifs increases, presumably due to dispersal of high-information content 'core' positions, which are typically adjacent in the real motifs. We, therefore, use and report results based only on null model 1. Sequence logos were constructed using the actual matches obtained in the ChIP-seq peak sequences, and the WebLogo 3.4 tool (*Crooks et al., 2004*). For each TF family *F*, we calculated an odds ratio (*OR*) comparing the ratio of families in CMs to the ratio of families in the motif list. We define *OR* as (a/b)/(c/d), where *a* is the number of TFs of family *F* involved in a CM; *b* is the total number of unique TF pairs involved in a CM, minus *a*; *c* is the number of TFs of family *F* in the motif list; and *d* is the total number of TFs in the motif list, minus *c*. We calculated the standard error as $\sqrt{(1/a + 1/b + 1/c + 1/d)}$, and the 95% confidence interval as $e^{ln(OR) \pm 1.96SE}$.

## Motif enrichment in co-regulated tissue/developmental stage-specific genes, KEGG and GO categories

We obtained selectively enriched gene sets for each tissue from (http://www.vanderbilt.edu/wormdoc/wormmap/), GO annotations from (http://www.geneontology.org/), and KEGG pathway modules (http://www.genome.jp/kegg/module.html). We ran FIMO (*Grant et al., 2011*) with default parameters.

## Data access

PBM-microarray data are available at GEO (www.ncbi.nlm.nih.gov/geo/) under accession number GSE65719. Motifs and 8-mer data (E- and Z-scores) are available at www.cisbp.ccbr.utoronto.ca. Supplementary data files, including plasmid and primer information, motifs, source data for figure heatmaps, and lists of TFs are found on our project web site http://hugheslab.ccbr.utoronto.ca/supplementary-data/CeMotifs/.

## Acknowledgements

This work was supported by CIHR Operating grants MOP-111007 and MOP-77721 to TRH SAL is supported by an Ontario Graduate Scholarship. HSN is supported by a CIHR Banting Fellowship. We are grateful to Andy Fraser, Susan Mango, Bob Waterston, Carlos Araya, and Don Moerman for helpful discussions and materials.

## Additional information

### Funding

| Funder | Grant reference | Author |
| --- | --- | --- |
| Canadian Institutes of Health Research | Operating grant MOP-111007 | Timothy R Hughes |
| Canadian Institutes of Health Research | Operating grant MOP-77721 | Timothy R Hughes |
| Canadian Institutes of Health Research | Banting Fellowship | Hamed S Najafabadi |
| Ontario Council on Graduate Studies, Council of Ontario Universities | Ontario Graduate Scholarship | Samuel A Lambert |

The funders had no role in study design, data collection and interpretation, or the decision to submit the work for publication.

## Author contributions
KN, SAL, MTW, Acquisition of data, Analysis and interpretation of data, Drafting or revising the article; AWHY, HZ, MA, Acquisition of data; JR, Analysis and interpretation of data, Drafting or revising the article; SM, Acquisition of data, Drafting or revising the article; HSN, Analysis and interpretation of data; JSR-H, JIFB, AJMW, Drafting or revising the article, Contributed unpublished essential data or reagents; TRH, Conception and design, Analysis and interpretation of data, Drafting or revising the article

# Additional files

## Supplementary files

• Supplementary file 1. Comparison of CisBP TF collection with wTF2.0. Includes comments of overlaps and differences between two lists and whether each entry is likely a *bona fide* TF.

• Supplementary file 2. *C. elegans* curated motif collection. This spreadsheet contains the curated motif IDs for each *C. elegans* TF along with their source and experimental support.

• Supplementary file 3. Number of experiments required for complete coverage of human, fly, and worm TF collections. This spreadsheet contains numbers of experiments needed for each DBD class to have complete coverage of the motif collection based on previously described DBD prediction thresholds.

• Supplementary file 4. List of primers and gene synthesis constructs used to obtain TF clones in this study. This spreadsheet contains primers used to clone TFs as well as gene synthesis constructs that were cloned in to the PBM plasmid backbone (*Supplementary file 5*).

• Supplementary file 5. PBM plasmid (pTH6838) backbone map. Information on the expression vector used in PBM experiments.

## Major datasets

The following dataset was generated:

| Author(s) | Year | Dataset title | Dataset ID and/or URL | Database, license, and accessibility information |
|---|---|---|---|---|
| Narasimhan K, Lambert SA, Yang AW, Riddell J, Mnaimneh S, Zheng H, Albu M, Najafabadi HS, Reece-Hoyes J, Fuxman Bass J, Walhout AJ, Weirauch MT, Hughes TR | 2015 | Mapping and Analysis of Caenorhabditis elegans Transcription Factor Binding Specificities | http://www.ncbi.nlm.nih.gov/geo/query/acc.cgi?acc=GSE65719 | Publicly available at the NCBI Gene Expression Omnibus (GSE65719). |

Standard used to collect data: MIAME.
The following previously published dataset was used:

| Author(s) | Year | Dataset title | Dataset ID and/or URL | Database, license, and accessibility information |
|---|---|---|---|---|
| Weirauch, *et al.*, | 2014 | Cis-BP | http://cisbp.ccbr.utoronto.ca/ | Publicly available at Cis-BP Database. |

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
