## [Decision Letter]

Thank you for sending your work entitled “Mapping and analysis of *Caenorhabditis elegans* transcription factor sequence specificities” for consideration at *eLife*. Your article has been favorably evaluated by Naama Barkai (Senior editor) and three reviewers, one of whom is a member of our Board of Reviewing Editors.

The Reviewing editor and the other reviewers discussed their comments before we reached this decision.

The authors describe a large-scale project to systematically characterize the DNA preferences of transcription factors in the *C. elegans* model organism. They use a well-established in-vitro method (PBMs) combined with validation again previous experimental results (both detailed and high-throughput assays). The authors provide a detailed comprehensive analysis of the data they collected. I commend them on the thoroughness of their investigation.

This manuscript presents a massive resource that I am sure will be useful for many researchers studying transcriptional regulation in *C. elegans*, and ones studying evolution of transcription factors and regulatory networks.

Several minor comments were raised that require your attention before publication:

In the Abstract, the authors should be more explicit about the number of TF for which a motifs has been determined directly for the first time thanks to this study: 71 old + (129 new-5 duplicate) + 97 predicted.

Figure 1, left panel: The use of a quadratic horizontal scale is non-standard potentially quite confusing since people expect a logarithmic scale but also see the zero. How about using a log scale but set zero to small positive value so that they are still visible?

Figure 1, right panel: These coverage values are quite meaningless, given the very small sample size for most families. We suggest removing this panel altogether.

Figure 2, panels A-C: The labels on the horizontal axis are unreadable. Use larger font.

Figure 4: The relationship between the colored heat map panel on the left of the amino-acid sequences and the black and white panel to the right of it is unclear. Also, why did the authors reverse the row order in the “blowup” panels on top and bottom compared to the full panels? This is unnecessarily confusing for the reader.

In the second paragraph of the Results section, we suggest the authors add “protein modifications (e.g. phosphorylation)” to the list of possible failures. The same could be added to the third paragraph of the Discussion section, and in fact protein modification is often used to control dimerization.

In the last paragraph of the subsection headed “Motif enrichment with Y1H and ChIP-seq data”: the sum of the different, mutually exclusive classes is larger than the total.

---

## [Author Response]

*In the Abstract, the authors should be more explicit about the number of TF for which a motifs has been determined directly for the first time thanks to this study: 71 old + (129 new-5 duplicate) + 97 predicted*.

We have now revised the Abstract to be more explicit about the number of motifs that have been determined directly, and the total number that are either measured or inferred. Because the number of words is restricted to 150, we do not explain the five duplicates—this is a relatively minor point, and it is made clear in the text—,and to reduce confusion, we also state that we can predict “many” rather than “97”.

Figure 1*, left panel: The use of a quadratic horizontal scale is non-standard potentially quite confusing since people expect a logarithmic scale but also see the zero. How about using a log scale but set zero to small positive value so that they are still visible?*

We have revised Figure 1 as the referee recommends. The left panel is now presented on log_2_ scale.

Figure 1*, right panel: These coverage values are quite meaningless, given the very small sample size for most families. I suggest removing this panel altogether*.

We have revised Figure 1 as the referee recommends. This panel has been removed.

Figure 2*, panels A-C: The labels on the horizontal axis are unreadable. Use larger font*.

We have increased the font size to match that of the rest of the sub-panels. This operation required removing some of the labels for the X-axis values, but they are easily interpolated and are explained in the figure legend.

Figure 4*: The relationship between the colored heat map panel on the left of the amino-acid sequences and the black and white panel to the right of it is unclear. Also, why did the authors reverse the row order in the “blowup” panels on top and bottom compared to the full panels? This is unnecessarily confusing for the reader*.

We have updated Figure 4 and its legend to address these issues. The black and white panel represents clusters of motifs, and was intended to illustrate that these clusters do not correspond closely to the sequence identities of the corresponding proteins (which are shown on the left), and the heat maps are intended to illustrate that the 8-mer data is more complex than the motifs can capture. We have revised the Figure 4, the Figure 4 legend, and the main text in an effort to clarify these points (see modifications to the second paragraph of the subsection entitled “Complex relationships between protein sequences and motifs recognized by the NHR family”).

To make the figure less complicated, we now highlight groups of related motifs on the dendrogram, using shading. This change is explained in the legend.

The reversal of row order in the blowup panels was unintentional and has now been corrected. We have also added NHR-204 to the bottom panel, which was also omitted in error. We apologize for these embarrassing errors.

*In the second paragraph of the Results section, we suggest the authors add “protein modifications (e.g. phosphorylation)” to the list of possible failures. The same could be added to the third paragraph of the Discussion section, and in fact protein modification is often used to control dimerization*.

Done.

*In the last paragraph of the subsection headed “Motif enrichment with Y1H and ChIP-seq data”: the sum of the different, mutually exclusive classes is larger than the total*.

We apologize that the text describing Figure 8 was incorrect. Figure 8 is itself correct and the manuscript now contains corrected text to reflect the figure. The passage in question now reads: “Out of the 635,712 possible patterns we tested, both methods call 635,483 insignificant, both call 49 positive, and they disagree on 180 (Figure 8).”